# An AI-Enabled Stock Prediction Platform Combining News and Social Sensing with Financial Statements

**Traianos-Ioannis Theodorou** [1,*], **Alexandros Zamichos** [1], **Michalis Skoumperdis** [1], **Anna Kougioumtzidou** [1], **Kalliopi Tsolaki** [1], **Dimitris Papadopoulos** [1], **Thanasis Patsios** [2], **George Papanikolaou** [2], **Athanasios Konstantinidis** [3], **Anastasios Drosou** [1] and **Dimitrios Tzovaras** [1]

[1] Centre for Research and Technology Hellas, Information Technologies Institute, 57001 Thessaloniki, Greece; zamihos@iti.gr (A.Z.); skoumpmi@iti.gr (M.S.); annak@iti.gr (A.K.); ktsolaki@iti.gr (K.T.); dpapadop@iti.gr (D.P.); drosou@iti.gr (A.D.); dimitrios.tzovaras@iti.gr (D.T.)
[2] Media2Day Publishing S.A., 15232 Athens, Greece; patsios@media2day.gr (T.P.); gpap@media2day.gr (G.P.)
[3] Department of Electrical Engineering, Imperial College London, London SW7 2AZ, UK; a.konstantinidis16@imperial.ac.uk
[*] Correspondence: theodorou@iti.gr

**Abstract:** In recent years, the area of financial forecasting has attracted high interest due to the emergence of huge data volumes (big data) and the advent of more powerful modeling techniques such as deep learning. To generate the financial forecasts, systems are developed that combine methods from various scientific fields, such as information retrieval, natural language processing and deep learning. In this paper, we present ASPENDYS, a supportive platform for investors that combines various methods from the aforementioned scientific fields aiming to facilitate the management and the decision making of investment actions through personalized recommendations. To accomplish that, the system takes into account both financial data and textual data from news websites and the social networks Twitter and Stocktwits. The financial data are processed using methods of technical analysis and machine learning, while the textual data are analyzed regarding their reliability and then their sentiments towards an investment. As an outcome, investment signals are generated based on the financial data analysis and the sensing of the general sentiment towards a certain investment and are finally recommended to the investors.

**Keywords:** Web 3.0; machine learning; sentiment analysis; portfolio optimization; portfolio management; media industry; social media; model-based trading





## 1. Introduction

The financial services sector provides services in the field of finance to individuals and corporations. This part of the economy consists of a variety of financial firms including finance companies, banks, investment houses and insurance companies. The industry of financial services is probably the most important sector of the economy, leading the world in terms of earnings and equity market capitalization.

Individuals may access financial markets such as stocks and bonds through investment services. Brokers that are either individuals or online services facilitate the buying and selling of securities. Financial advisors are responsible for the assets management as well as the trades of the portfolio assets. Additionally, quant fund is the latest trend of financial advice and portfolio management, with fully automated algorithmic portfolio optimization and trade executions. Quant funds refer to investment funds whose securities are chosen based on the quantitative analysis of numerical data. The analysis is performed by software programs which utilize advanced mathematical and Artificial Intelligence (AI) models. These models deal with several challenging tasks in the financial services sector, such as the accurate prediction of the stock prices. In contrast with traditional methods, quant funds rely on algorithmic or systematically programmed investment strategies. Thus, they

are not human-oriented and their decisions rely only on data analysis. In this way, they can be very useful tools for supporting investors in their investment's decision making. Regarding the stock prices prediction, many methods have been proposed by researchers since the beginning of the stock market [1–7].

Moreover, during the last years with the advent of big data and machine learning in Web 3.0 technologies, financial forecasting and model-based trading have gained growing interest. Web 3.0 is the third generation of Internet services for websites and applications that focus on using a machine-based understanding of data to provide a data-driven and semantic web. While the traditional methods for financial forecasting were based on the analysis of the stock prices of the past, nowadays, the efforts consider a variety of data. More specifically, the huge amount of available data can be found on social media and from the media industry (articles and posts) that may be related with significant societal and political events. These events, as well as the extracted information from their sentiment analysis, may reflect the stock prices and trends of certain assets. Additionally, investors tend to follow the general sentiment on a specific topic, something that affects their final decisions. However, the use of machine learning algorithms and the large amount of structured data that is produced on a daily basis can give them a more clear and personalized view of the sentiment and trend of their portfolio assets, assisting their final decisions.

In this work, we propose ASPENDYS, an interactive web-platform that offers supportive information to investors. The primary aim of the ASPENDYS platform is to assist investors in the decision making related with stock investments, taking into consideration multiple sources and a large amount of data by using state of the art prediction algorithms. Initially, in Section 2, we review the related literature and defined the research gap and question that our work needs to cover and answer respectively. Then, in Section 3, we describe the methodology, architecture, data, components and functionalities developed in the ASPENDYS project. More specifically, the research methodology is described in Section 3.1 and the architecture of the platform is analyzed in Section 3.2. In Section 3.3, we describe all the data used by the algorithms of the components. Section 3.4 describes the algorithms that were used for the extraction of both articles sentiments and the asset sentiments. Additionally, Section 3.5 analyzes the methodology that was used in the production of the data reliability metric. Moreover Section 3.6 describes the methodologies that are supported for the optimization of the user portfolio. Finally, Section 3.7 presents the different investment signals generators that are developed and used in the ASPENDYS platform. In Section 4, we describe the resulting user interface of the ASPENDYS platform, as well as two application use cases. Additionally, in Section 5, we present the summary of this work and in Section 6 we analyze how we answered the research questions as well as the limitations of our study.

## 2. Related Work

In this section, a detailed literature review is presented regarding stock prediction and sentiment analysis methods. The section consists of three subsections. Initially, the related works are divided into two eras, the "before AI era" and the "AI era". Moreover, works that justify the correlation of the stock predictions and the general sentiment extracted from social media are presented. Following this, the academic gaps, the research questions that arise and the contribution of the proposed platform are defined.

### 2.1. Before AI Stock Predictions Solutions

The first research attempts in the field of stock prediction values are based either on econometric models or on technical indexes. The most popular statistical models, which are widely utilized in econometrics for future prices prediction, are ARIMA and ARMA [2]. Two popular indexes are utilized for stock assets price prediction: Moving Average Convergence Divergence (MACD) and Relative Strength Index (RSI). The Volatility of Average Convergence Divergence [8] is found to be effective for the prediction of a

specific asset [8]. Moreover, both indexes are proved to be more efficient in stock market trends prediction than SELL and BUY strategies [9]. Nowadays, state-of-the-art methods are based mainly on machine learning and deep learning algorithms that have been found that they achieve better results in stock market prediction.

Moreover, the stock prediction can be enhanced by, or even solely based on, market information that can be derived from the news spread and circulated in the societal sphere. Numerous studies have been conducted with the aim to capitalize on this available information, particularly since it can be accessed conveniently via digital media. Thus, the field of sentiment analysis, and the closely related opinion-mining field, emerges. The subject of the sentiment analysis is the automatic extraction of sentiments, evaluations, attitudes, emotions and opinion. The emergence and growth of these field takes place in tandem with the rise of the social media on the web, e.g., reviews, forum discussions, blogs, micro-blogs, Twitter and social networks, and is clearly related to the fact that a huge volume of data containing opinions is recorded and available. Sentiment analysis has evolved to be one of the most active research areas in natural language processing. Data mining, web mining and text mining also employ sentiment analysis. Apart from being a computer science field, it has expanded to management sciences and social sciences due to its importance to business and society and sentiment analysis systems have found their applications in almost every business and social domain [10,11]. The stock market domain of course could not constitute an exception.

Two main approaches to the problem of extracting sentiment automatically, prior to the emergence of AI, can be identified: the lexicon-based approach for automatic sentiment analysis and the statistical classifier approach. The lexicon-based approach involves calculating a sentiment for a document from the semantic orientation of words or phrases in the document. The lexicons contain words or multiword terms tagged as positive, negative or neutral (sometimes with a value reflecting the sentiment strength or intensity). Examples of such lexicons include the Hu & Liu Opinion Lexicon, the SentiWordNet Lexicon, the Multi-perspective Question Answering (MPQA) Subjectivity Lexicon, the General Inquirer, the National Research Council Canada (NRC) Word-Sentiment Association Lexicon and the Semantic Orientation Calculator (SO-CAL). The statistical text classification approach involves building classifiers from labeled instances of texts or sentences, essentially a supervised classification task [11–13]. Most notably, two lexicon-based mood tracking tools, OpinionFinder that measures positive versus negative mood and Google-Profile of Mood States that measures mood in terms of six dimensions, were used in a context related to the scope of the present work to investigate the hypothesis that public mood states are predictive of changes in DJIA closing values [14].

Issues regarding lexicon-based methods include the fact that the dictionaries are deemed to be unreliable, as they are either built automatically or hand-ranked by humans [11], as well as the need for the lexicons to be domain specific. Moreover, given a sufficiently large training corpus, a machine learning model is expected to outperform a lexicon-based model [13].

### 2.2. AI Era Stock Predictions Solutions

Future price prediction is a complicated task in machine learning field as it is not clear if asset prices embody investors' behavior or stock prices time series are random walks. In the literature, various AI-based and state-of-the-art studies attempted to predict future prices [3–5]. A variant of Neuro-fuzzy system which employs both recurrent network and momentum is utilized for the prediction of four assets of Dhaka stock exchange [3]. Various machine and deep learning methods are used, such as K-Nearest Neighbor regression, Generalized Regression Neural Networks (GRNN), Support Vector Regression (SVR), Multi-Layer Perceptron (MLP), Recurrent Neural Network (RNN), Gaussian Processes and Long-short term memory (LSTM) to predict future values of 111 stock market assets [4]. The employment of Wavenet model to predict the S&P 500 future prices achieves lower Mean Absolute Scaled Error (MASE) compared with the aforementioned methods [5]. The

effectiveness of Wavenet model is the main motivation to be utilized in the prediction of assets' future values in platform.

With the emergence of AI-based techniques, deep learning neural networks became the dominant approach to tackle natural language and text processing. These deep learning approaches include many networks types such as Fuzzy Neural Networks, Convolutional Neural Networks (CNNs), Recurrent Neural Networks (RNNs) and more recently transformer-based neural networks [15–17]. Numerous studies have been conducted aiming to leverage the great amounts of available data to extract more accurate and exploitable stock market information. Sentiment analysis and machine learning principles were applied to find the correlation between "public sentiment" and "market sentiment"; firstly, Twitter data were used to predict public mood and subsequently stock market movement predictions based on the predicted mood and the previous days' DJIA values were performed [15]. In a later work, sentiments were derived from Yahoo! Finance Message Board texts related to a specific topic of a company (product, service, dividend, etc), employing thus the "topic sentiment" and not the "public sentiment". The sentiments were incorporated into the stock prediction model along with historical prices [18]. Sentiment analysis and supervised machine learning principles were also applied to the tweets about a company and the correlation between stock market movements of the company and the sentiments in tweets was analyzed, concluding that a strong correlation exists between the rise and fall in stock prices with the sentiments expressed in the tweets [19]. In [6], the proposed prediction model was based on sentiment analysis of financial news and historical stock market prices. In the scope of the specification of trading strategies, taking into account the decaying effect of news sentiment, thus deriving the impact of aggregated news events for a given asset, was also proposed [20]. Trading strategies that utilize textual news along with the historic prices in the form of the momentum to obtain profits were designed by Feuerriegel and Prendinger [7].

Many works have been published concerning financial forecasting and portfolio management platforms utilizing AI technologies. Researchers are addressing the challenges arising from the application of AI technologies in quantitative investment. In particular, they have designed and developed an AI-oriented Quantitative Investment Platform, called Qlib. It is based on attempts to build a research workflow for quantitative researchers, utilizing the potential of AI technologies. Their platform integrates a high-performance data infrastructure with machine learning tools dedicated for quantitative investment scenarios. It provides flexible research workflow modules with pre-defined implementation choices regarding investment, as well as a database dedicated to scientific processing of financial data, typical for tasks in quantitative investment research. Finally, by including basic machine learning models with some hyper-parameter optimization tools, the authors reported that Qlib outperforms many of the existing solutions [21].

There are recommendation systems supporting investment decisions, which are based on extracting buy/sell signals retrieved from technical analyses and predictions of machine learning algorithms. Through the platform's interface, users are able to derive their own conclusions and manage their investment decisions by evaluating the analyzed information by the system. The authors' development was based on retrieving historical financial data regarding various financial products: stocks, funds, government bonds, certificates, etc. The predictions on the future behavior of a product were derived through machine learning regression algorithms, such as Random Forest, Gradient Boosting, MLP and KNNeighbors. The technical analysis factors were calculated by utilizing indicators such as Relative Strength Index (RSI), Stochastic Oscillator (STOCH), Simple Moving Average (SMA), Exponential Moving Average (EMA) and others. Finally, their platform provides representations of prices' time-series, tables and charts including financial products' open-high-low-close values that depend on the aforementioned analyses [22].

Some works integrate sentiment analysis with a SVM-based machine learning method for forecasting stock market trends [23]. Their contribution consists of considering the day-of-week effect to derive more reliable sentiment indexes. The financial anomaly day-

of-week effect [24], which indicates that the average return on Mondays is much lower than that on the other weekdays, influences the accuracy of the investor sentiment indexes. Researchers are using this effect by introducing an exponential function on past sentiment changes on weekends and then generalize to holidays, to adjust the sentiment indexes. To forecast stock market movement direction, they extract features from unstructured financial review textual documents from which the sentiment indexes are constructed. Lastly, a SVM model is employed to produce the stock price predictions by implementing five-fold cross validation and a rolling window approach.

Some works employed sentiment analysis on Twitter posts in an attempt to produce social signals to be used along with historical market data, in order to predict cryptocurrency prices. They tackled the sentiment analysis task using VADER, and, along with features produced from market data, they trained MLP, SVM and RF algorithms [25].

There are works that used stock price historical data in combination with financial news articles' information to predict the prices of stocks. Testing the performance of ARIMA, Facebook Prophet and RNN models without textual information, as well as the performance of RNNs with additional textual information, they concluded that the latter performed better. More precisely, their results show that an RNN using price information combined with textual polarity calculated by the nltk Python library, in most cases performed better than the aforementioned models as well as than that of an RNN operating with price data along with the text as inputs [26].

### 2.3. Defining the Academic Gaps and Research Questions

Based on the aforementioned analysis, it is concluded that there are various methods in the literature that deal with the stock prediction. These methods utilize either statistical methods or AI methods including DNNs. Most of the presented methods use historical values of the stock prices in order to predict the forthcoming prices, while there are others that try to predict stock prices through sensing the general sentiment on social media.

Even though there are many works dealing with stock prediction, there are still some gaps that have to be considered. First, for the stock prediction, most of the existing works are based on stocks' historical values. This approach lacks of considering other information that may be significant, such as social or political events that are depicted in news articles and social media posts. Such information may lead to the early detection of critical changes in the trends of stock prices. Feuerriegel and Prendinger [7] proved that such information can be considered by sensing public sentiment on social media. Thus, the monitoring and analysis of social medias' sentiment can be of added value for the stock prediction. In this aspect, there is a lack of platforms that handle and combine both historical stock prices and the general sentiment for the investment recommendation. Second, although there are many platforms or systems that provide advice for investments, also called robo-advisors, they usually act automatically without providing to the investors the information that led to their recommendations. Thus, the interpretability of their recommendations is not well established. Finally, there is the need for an interactive platform that will collect and analyze data in a daily basis and provide investment recommendations not only for specific assets but for an investor's portfolio. The suggestions should be personalized to the profile of the users so that different advice can be provided to risky and conservative investors.

Following the above research gaps, we propose ASPENDYS, an interactive web-platform that offers supportive information to investors, thus facilitating the decision making related with stock investments. The platform consists of several tools and methods that utilize key technologies of the computer science. These include Natural Language Processing (NLP), a sub-field of computer science, and artificial intelligence that can accurately extract information and insights contained in the textual data as well as categorize and organize these data. Additionally, in the ASPENDYS platform, deep learning (DL) technologies were used, which is a sub-field of machine learning concerned with algorithms inspired by the structure and function of the brain called artificial neural networks.

The research questions (RQ) that have arisen are as follows:

RQ1: Is the ASPENDYS platform able to provide investment recommendations by combining both historical stock prices and the extracted sentiment, for an asset, through the sensing of social media and news websites?

RQ2: Can ASPENDYS platform be used as a supportive tool for investors that will assist their investment decision making and monitoring?

## 3. Materials and Methods

### 3.1. Methodology

Initially, the work started with the collection of the requirements from real users. For this purpose, a questionnaire was created that was shared to 30 individuals of the investment private sector in Greece. The end users became aware of the scope of the ASPENDYS project and the goals of the systems that is going to be developed before filling in the questionnaire. The results of this research led to the definition of the specification of the ASPENDYS platform as well as its architecture. More specifically, we concluded that there was a necessity for implementing components that will collect data from different sources, check the reliability of these data, generate investment signals and optimize the user portfolio. All of these components should be integrated into a web application with a friendly user interface. Following that, we defined the system architecture and declared the connections among the platform components as it is described in the following sections. The next step in our research was the literature review for the specific research field in order to identify the state of the art in the stock market prediction, as described in detail in the previous section. Based of this literature review, we started implementing each of the architecture components. Moreover, we defined the sources of both textual and historical stock prices. Regarding the textual data, we defined that we should use news websites in order to collect news articles that regard certain assets. In addition, for the collection of information from social media, Twitter and Stocktwits were selected. Twitter is one of the most popular social media sites where users express their opinion on various topics. In addition, they utilize the functionality of hashtags, which allows the filtering and retrieval of information in an efficient way. Moreover, Stocktwits was selected, as it is very similar to Twitter and its users are mainly investors. Regarding the retrieval of the stock prices, Yahoo! Finance was selected, since it provides both historical and up-to-date stock prices that are updated daily. The final step of our work was the integration of the modules to a unique platform that exports each component service through RESTful APIs to the ASPENDYS web application as well as to run several use cases in order to validate our system, which is described in Section 4.

### 3.2. System Architecture

In Figure 1, the architecture of the ASPENDYS platform is illustrated. The platform is separated into seven modules: Data Collection, Database Management System, Portfolio Optimization, Sentiment Analysis, User Portfolio Management and User Interface.

The Data Collection module is responsible for the extraction of data from various data sources. It consists of two sub-modules: one retrieves data from the social media using the Twitter and Stocktwits API, and the other extracts articles and financial data using the News API and the Yahoo! Finance API. The Database Management System is responsible for storing and retrieving the data from news articles and financial indicators that are collected from the Data Collection module. Additionally, this module is used for user management, for both registration and user data retrieval. The DBMS that is used is a MySQL server, and the application logic is implemented in Python. The Portfolio Optimization module is triggered by the User Interface, and its results are shown on it. This module proposes a better synthesis of the portfolio based on different methods, which are analyzed in the following subsection.

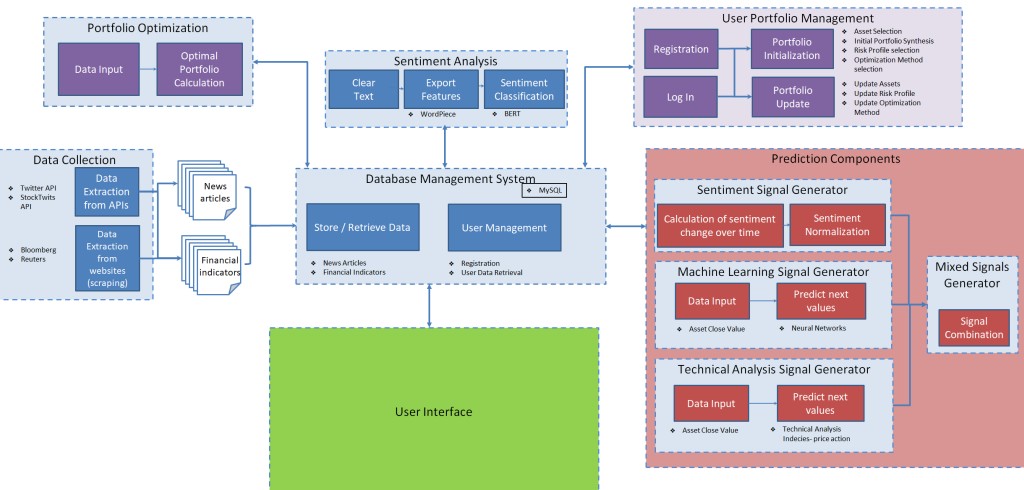

**Figure 1.** ASPENDYS platform architecture.

The Sentiment Analysis module uses input from the Data Collection module through the database in order to produce the sentiment of the articles and the respective assets. Using the User Portfolio Management module, the user of the platform is able to create a new portfolio as well as update the current one. The interaction with this module can be done through the User Interface and the results are stored in the Database Management System. The Signal Generators module consists of four sub-modules that generate investment signals using different input data. These sub-modules are sentiment, machine learning, technical analysis and mixed signal generators; more details are provided in the following sections. The User Interface is the front-end of the ASPENDYS platform where the user is able to visualize the data in graphs and tables, as well as manage their portfolio. It is an Angular application [27] that communicates with the data management system as well as the portfolio optimization and the user portfolio management module.

### 3.3. Data Collection

In this section, the data collection processing of both texts and prices is described. Texts are sorted in short-text data derived from Twitter and Stocktwits and long-text data derived from financial news article platforms, whereas stock prices come from the Yahoo! Finance site. All data, both textual and numerical, are stored and integrated in SQL type database. In next subsections, data collection processing and data features are presented in detail.

### 3.3.1. Twitter

To collect tweets from the social media platform Twitter, we used tweepy [28], an open-source Python library, which handles the connection to Twitter's API [29] given the required account information and streams tweets based on certain word filters and other search parameters. On a daily basis, the platform collects tweets for each asset, based on the condition of including the asset's name or using a set of relevant hashtags. The information stored for each tweet is tweet's id, text, number of re-tweets, creation date, user's screen name, whether the user is verified, user creation date, number of followers, status count, number of friends and, finally, number of mentions, URLs and hashtags used. A tweet is stored in the database, after checking its credibility and if its id (incremental integer value) is larger than the most recently stored tweet in the database, ensuring that no duplicates are stored. The credibility of the tweets is checked by classifying them into two classes (credible or non-credible) by a Random Forest classifier trained on the CREDBANK dataset [30] and based on the rationale described in [31]. The features of tweets used as input for the classifier are: the user creation date, whether the user is verified, the status count and the numbers of followers, friends, mentions and URLs and hashtags used.

### 3.3.2. Stocktwits

Regarding the collection of data from Stocktwits social media platform, we used Stocktwits' API [32]. Stocktwits for each asset are retrieved daily using the asset's ticker. The information stored for each one is the id, text, number of mentioned users, creation date, sentiment of the stocktwit as generated by the platform, username, user creation date (join date), whether the account is official, number of followers and number of accounts that the user is following. Consequently, the credibility of the stocktwits is checked, and they are stored in the database if their id is larger, meaning they are more recent than the last Stocktwits post in the database. The credibility of the stocktwits is checked using the same Random Forest classifier used in the case of tweets retrieved from Twitter, the only difference being the features used as input for the classifier. Since less information is available for these stocktwits, only the creation data, whether the account is official and the numbers of followers and accounts that the user is following are used as input for the classifier, with all other count features regarded as equal to zero.

### 3.3.3. News Articles

To perform sentiment analysis on news data, concerning the involved assets, we collect news articles on a daily basis from six liable sources: BBC News, Reuters, Coin-telegraph, Yahoo! Finance, The Wall Street Journal and Bloomberg. The search terms applied to retrieve relevant articles is the asset's name, as well as the asset's ticker. For some assets, only the asset's name was used as a search term, as their ticker is a too vague term (e.g., C for Citigroup, V for Visa or F for Ford). For the articles' collection, a set of web scrapers were developed and used in combination with the Python library NewsAPI [33].

The following information is collected for every gathered news article: the article's title, the full text, posting date, writer and source's website. Subsequently, the collected articles are checked to not have a very similar body with each other, as well as with every article in the database. To achieve this, the text similarity between each pair of articles is calculated, utilizing the NLP Python library Spacy [34]. The motivation behind this is to avoid storing and analyzing the same article multiple times, as particularly similar articles could indicate that the same article has been retrieved multiple times from different sources or that the same article with minor changes has been re-posted. In both cases, very similar articles need to be filtered out.

### 3.3.4. Assets Values

For the collection of the assets values, the Python module "pandas datareader" [35] was used, in order to extract data from the Yahoo! Finance API. More specifically, it collects open, close, high, low, volume and adjusted close values for each asset on a daily basis. The open value is the initial price of each asset when a trading session starts and the close value is the last exchange price at the expiration of a trading session. The adjusted close value amends an asset's close price to reflect that asset's value after accounting for any corporate actions. The high value is the highest selling price of the asset and the low value is the lowest selling price of the asset during trading hours. The volume value is the total number of shares exchanged between stakeholders during trading hours multiplied by the current selling price of the asset.

### *3.4. Sentiment Analysis*

In this paper, we apply text sentiment analysis to extract the sentiment of stock-related texts, namely articles, tweets and stocktwits. We aim to capture the sentiment regarding a specific asset as expressed in a number of texts whose reliability we have previously confirmed.

### 3.4.1. Articles and Tweets Sentiment Method

We perform sentiment classification per article, tweet or stocktwit on 5-point scale ranging from $-2$ to $2$. The scores $[-2, -1, 0, 1, 2]$ correspond to the five different classes

[very negative, negative, neutral, positive, very positive]. (Through the UI, the user has access to the sentiment value attributed to each article, tweet or stocktwit, as well as the text of the article itself).

For the classification task, we used the neural network transformer-based model, Bidirectional Encoder Representations from Transformers (BERT) [17]. BERT is a language model for Natural Language Processing (NLP) that has achieved state-of-the-art results in a wide variety of NLP tasks. To adapt BERT to our specific task, we fine-tuned it on the Stanford Sentiment Treebank-5 (SST-5) dataset [36] for fine-grained sentiment classification. The fine-tuned model accepts as input the text and outputs the class ($[-2, -1, 0, 1, 2]$) attributed to text providing thus an estimation of the sentiment best describing the text.

### 3.4.2. Asset Sentiment Method

We approach the task of estimating the total sentiment corresponding to a specific asset in the present moment by equally weighting the asset sentiment derived from the most recent, newly sourced, texts and the averaged sum of the previous decayed asset sentiment values and computing their average. For the computation of the asset sentiment from the recent texts, we first aggregate all the recently acquired and not processed yet texts (articles, tweets and stocktwits) related to the specific asset, and then, using the model described above, we attribute sentiment values to each of the texts and after that we average the values.

### 3.5. Author and Source Reliability

Author and source reliability is the process where an indication of the level of trustworthiness of an article author or of a news feed site is extracted. This module is not related with social media platform post reliability but it does correspond with news media articles reliability. Unless news media articles are considered legit and reliable, ASPENDYS platform receives numerous articles from various sources so the extraction of their reliability is useful information. This task is complicated because there are no objective criteria for the reliability of an author or a media source according to recent studies. More specifically, the annotation of news articles is required to be performed by humans to classify them as either reliable or unreliable [37,38]. This process has been found to be biased since humans proved to be affected by the popularity of the author or news site [38,39]. Additionally, in some cases, it is taken for granted that textual data are labeled as written by a reliable author or not [40]. It is remarkable that state-of-the-art methods focus either on fake news detection or on author reliability rather than on the employment of a global method resolving both of these issues. In this study, based on the intuition that reliable level is not absolute but relative, both author and source reliability are estimated based on similarity with the rest of all of the items. More specifically, a profile for each author and source is generated and for each these items the average similarity with the others is computed; the item with the highest similarity is assigned a predefined high score; and, lastly, for the rest of items, values proportional to the highest value are assigned. Generally, in the case of author and source reliability, the more similar a profile is to the others, the higher is the reliability score assigned to it. To estimate the similarity between items, we utilize the function words for each sentence [41]. Function words were found to be content-independent and efficient in the extraction of writing profile of authors. More specifically, they belong to a specific set of parts of speech such as prepositions, conjunctions and pronouns. Each function word is relative to either itself or another function word if they are inside the same sentence provided they are contained in the processing window. The assigned score for each word is based on both adjacency network and Markov chain model, as its relation with another word decreases as their distance increases. For the comparison of the writing profiles of authors and sources, Kullback–Leibler divergence is utilized.

*3.6. Portfolio Optimization*

Portfolio Optimization is the process that determines the distribution of stock market products that make up a portfolio based on certain sizes. These figures are usually the expected profit and risk, always based on the assumption that the risk is intertwined and proportional to the profit. The real revolution in portfolio management came in 1952 from Harry Markowitz (Nobel Prize in Economics in 1990), who introduced and built what became known as Modern Portfolio Theory (MTP). Modern Portfolio Theory is aimed at investors. It applies the way in which we can optimize/maximize the expected return of a portfolio given the level of risk we want to take. Similarly, given the desired performance of a portfolio, we can build a portfolio with the least possible risk [42,43]. Regarding the current project and for the Portfolio Optimization of the users, we applied the follow methodologies:

- Portfolio Optimization based on Modern Portfolio Theory [44] is based on fundamental concepts of Statistics such as Variance and Correlation. It proves that it is more important to measure the performance of a portfolio by the total portfolio of products that make it up. The set of Optimized Portfolios constitutes a curve called the Efficient Frontier. Thus, with each desired expected return (Y-Return axis), we are able to know our Optimized Portfolio that results from the corresponding point in the curve and of course corresponds to the minimum risk. Similarly, for each level of risk we are willing to take (X axis—Standard Deviation (Risk)), we look for the corresponding point in the Effective Front curve, which returns us the Optimized Portfolio with the maximum possible return.
- Portfolio Optimization based on Modern Portfolio Theory is the selection of the minimum variance-risk (Minimum Variance Portfolio), which is a variation of the Modern Portfolio Theory. From the Efficient Frontier curve, we choose the point that constitutes a diversified Optimized Portfolio consisting of stock products that provide the least possible risk.
- Portfolio Optimization Based on the Black–Litterman Model is a mathematical Portfolio Optimization model developed in 1990 at Goldman Sachs by Fischer Black and Robert Litterman and published in a more enriched version in 1992 in the Financial Analysts Journal. This model solves some problems faced by institutional investors but also ordinary traders when they apply the Modern Portfolio Theory in practice. The innovation of Black–Litterman model is that the investors are able to define certain views for each one of the assets of the portfolio [45–47].
- Portfolio Optimization based on the Risk Parity model is a Portfolio Optimization strategy used extensively by many investment schemes (mainly hedge funds) for maximum return, given a level of risk through the portfolio risk method (Equally-weighted risk contributions portfolio). Each product participates in the Portfolio in the same way as in the total variation of the Portfolio [48,49].

*3.7. Signals Generators*

In the proposed platform, we developed four different investment signals generators. More specifically, the generator based on the asset sentiment, the machine learning generator, the technical analysis generator and the mixed generator. These generators produce two types of signals: the SELL signal that proposes to the user to sell the asset and the BUY signal that proposes to user either to increase the percentage of investment in a specific asset or to buy this asset if it is not included in their portfolio. In the following subsections, we describe in details these components.

3.7.1. Sentiment Signal Generator

The method (strategy) we implemented for the sentiment signal generation utilizes the (total) asset sentiment along with the historic prices in the form of the momentum. The rationale behind this choice is that we want to invest in assets with both a high polarity sentiment and previous momentum in the same direction. Thus, trading signals based on sentiment are generated only if both the asset sentiment and the asset's historic prices

provide an evidence of continuation in the same direction. Regarding the sentiment analysis expressed in the articles and the social media posts, the fact that there was no available dataset containing financial texts annotated with fine-grained sentiment labels constituted a challenge. For this reason, the Stanford Sentiment Treebank-5 (SST-5) that contains reviews from the Internet Movie Database (IMdb) was used to fine-tune our model, based on the fact that is constitutes a reliable dataset with fine-grained (5 classes) sentiment annotations.

### 3.7.2. Machine Learning Signals

Financial Signals are based on predicted close values, which are formed to belong to one of the following categories: BUY and SELL. For the extraction of the predicted values, a deep learning method that is based on a modified version of WaveNet model [50] was found to be effective in the multivariate forecasting of financial time series [5].

The prediction of time series is implemented under the condition of other highly correlated time series based on Wavenet model standards. The WaveNet model [50] is implemented with the application of a dilated convolution neural network to distinguish the predicted prices from the actual prices, instead of utilizing the recurrent connections in which the preceding values are taken into account. In the dilated convolution process, a filter is applied in a more extended area than its length and the input values are skipped by a stable step.

The architecture of the method combines one convolutional layer and two dense layers with Relu activation function. The hyper-parameters of the model were set as follows: the number of kernels equal to 64, the kernel size equal to 2, the time-step equal to 1 and the processing window equal to 64. The input vector is based on 60 financial indices found to be effective in stock price forecasting [51] and is estimated by the combination of open, close, low, high, volume and adjusted close values of market stocks. Input data are initially preprocessed by applying the min-max normalization method on them.

Feature selection is based on method developed by Yuan et al. [52]. More specifically, the Random Forest feature selection algorithm is applied on the input data. A financial index considered more important when it increases the difference between the out of bag score in a feature vector with added random noise and the out of bag score in the initial feature vector. Finally, 44 financial indices out of 60 are selected. The post-processing of predicted values is defined as their transformation to financial signals, so each prediction should be mapped to one of the two aforementioned categories. For this purpose, each prediction is transformed to its percentage variation in relation with the preceding prediction value to be estimated as the return values of each asset. In the case of the return value is higher than double the summation of average and standard deviation of the return distribution, the extracted signal is equal to BUY. When the return value is lower than double the standard deviation minus the average of return distribution, the signal is assigned to be equal to SELL.

### 3.7.3. Technical Analysis

Technical Analysis is an investment and trading tool that spots investing opportunities and produces trading signals by analyzing statistics that come from the traders and markets' activity such as the price action and the trading volume. Technical Analysis focuses mainly on the study of the price and the volume of a trading asset.Additionally, the Technical Analysis is based on the Efficient Market Hypothesis, a hypothesis that states that the prices of stock market products reflect all the information that concerns them. Technical analysis also assumes that values move in iterative patterns and that these patterns are often repeated. The main problem that Technical Analysis faces is that it often contains subjectivity. Indicators, oscillators and other objective tools should be used; all technical analysts should see the same indicator and the same values, without being dependent on them. The portfolio optimization presents various challenges such as the fact that too many transactions are proposed. This is addressed by properly configuring the system and selecting low-cost changes. Minimal variation portfolios without the expected returns can

also be proposed. This can be resolved by minimizing the tracking error in the portfolio. At the same time, the mean variance optimization is restrictive. In this case, an optimization algorithm can be selected with a few limitations. As many times one of the products participating in the portfolio takes too much weight in the portfolio, , the selection of products should be based on specific rules to avoid this. The tools of Technical Analysis thoroughly study the ways of supply and demand of a trading asset and how these factors affect the price, trading volume and volatility. Technical Analysis is also used for the production of short-term trading signals through charting tools as well as a tool for the evaluation of certain trading assets regarding the relative markets or relative sectors [53,54]. Recently, many new developments have appeared in the Technical Analysis field including methodologies, algorithms, indicators, oscillators, etc. [55], but they are all based on three very simple assumptions [56]:

1.  Prices reflect anything: Most technical analysts believe that any factor that is correlated to a trading asset, which is tradable in the global markets, from the fundamentals up to the market psychology, is already reflected in its price. This assumption is aligned with the famous Efficient Market Hypothesis (EMH), which comes to the same conclusions regarding the prices. Therefore, what technical analysts suggest is price analysis, which, according to the aforementioned assumption, contains all of the information of the asset such as the supply and demand of a stock or a commodity.
2.  Price movements are trending: Technical analysts expect that the asset prices, even in random market movements, come out in trends, no matter what the observation timeframe is. The price of a stock, most likely, will continue following the trend, instead of moving randomly or erratically. Most Technical Analysis strategies contain trend as a basic feature.
3.  History tends to repeat itself: Technical analysts believe that history repeats itself. The repetition of price movements is often attributed to market psychology that tends to be quite predictable as it is based on emotions such as fear, over-optimism, panic, etc. Technical analysts use chart patterns to analyze emotions and possible subsequent movements in the markets to understand trends. In recent decades, countless trend-setting techniques have been developed, with different approaches and tools.

Regarding the current project, to extract trading signals, we used the following Technical Analysis tools and indicators:

- MACD Indicator (Moving Average Convergence-Divergence) to define the market trend.
- ADX Indicator (Average Directional Index) to define the market trend strength.
- High and low prices in specific timeframes to find supports and resistances (price levels that are difficult to breakout either in a bullish trend (resistance) or in a bearish trend (support).
- RSI Indicator (Relative Strength Index) to define any possible overbought/oversold price levels in order to spot any possible corrective movements or even a trend reversal.
- ATR Indicator (Average True Range) to define the market volatility in order to apply the optimum risk management as well as the stop loss and take profit levels.

Trading signals (BUY/SELL) are produced, by combining all the above indicators [53–58]:

- MACD Indicator gives the market trend. MACD > 0 means uptrend, which is a BUY signal, while MACD < 0 means downtrend, which is a SELL signal.
- ADX Indicator should be >25 in order to have enough market strength. It is a filter to both BUY/SELL signals.
- Current price should NOT be close to resistance for BUY signals or close to support for SELL signals. It is a filter for both BUY/SELL signals.
- The asset should NOT be in overbought condition for BUY signals or in oversold condition for SELL signals, as defined through RSI indicator.
- The suggested size of each signal is defined through ATR as well as the suggested take profit/stop loss levels.

### 3.7.4. Mixed Signal Generator

Machine learning, Sentiment Analysis and Technical Analysis modules consider each asset price as independent. In mixed signals module, the intuition that each asset behavior is affected by prices of other assets is taken into account. Signals of all modules are integrated and it is assumed that each asset close price prediction should be based on both close prices and signals of itself and others.

For this purpose, it is essential for close prices of assets to be transformed in financial signals. Based on the moving average rule described in work [59], the extracted mixed signals are compared to the values between long-run and short-run moving averages with a band value taken into account. More specifically, in the case that the long-run moving average is higher than the summation between the corresponding short-run and the band value, the BUY signal appears.

In contrast, a SELL signal is derived in the case where the long-run average is lower than the short-run counterpart minus the band value. Cases in which the difference between the long-run and short-run moving average is between 0 and the band value are not taken into consideration. The long-run moving average is defined as equal to 50, the corresponding short-run equal to 1 and the band equal to 0.01 as dictated by Gunasekarage and Power [59] as the optimal parameters combination to achieve maximum profit. To train the input data, the method described by Spyromitros-Xioufis et al. [60] is employed, which is an advanced version of logistic regression algorithm with the extension of a Lasso regularization term.

## 4. Results

The results of this research concern the presentation of the user interface functionality as well as some use cases of the presented platform.

### 4.1. Aspendys User Interface

The ASPENDYS User Interface is separated into two main views that contain widgets: the Portfolio Management view and the Investment Sentiment Analysis view. Both views share a common component which is the Investment Signals. In the following subsections, these resulting views are analyzed in detail.

### 4.1.1. Portfolio Management

The Portfolio Management view is one of the two main views of the ASPENDYS platform. It contains four unique widgets that assist the user to be informed about the process of their portfolio as well as to manage it: (1) the Portfolio Overview; (2) the Profit/Loss History; (3) the Signal Acceptance History; and (4) the Current Portfolio Synthesis. These four widgets are included in the specific view, as shown in Figure 2 and analyzed in this section. The Portfolio Management view also contains the widget Investment Signals that is common in both views of the platform.

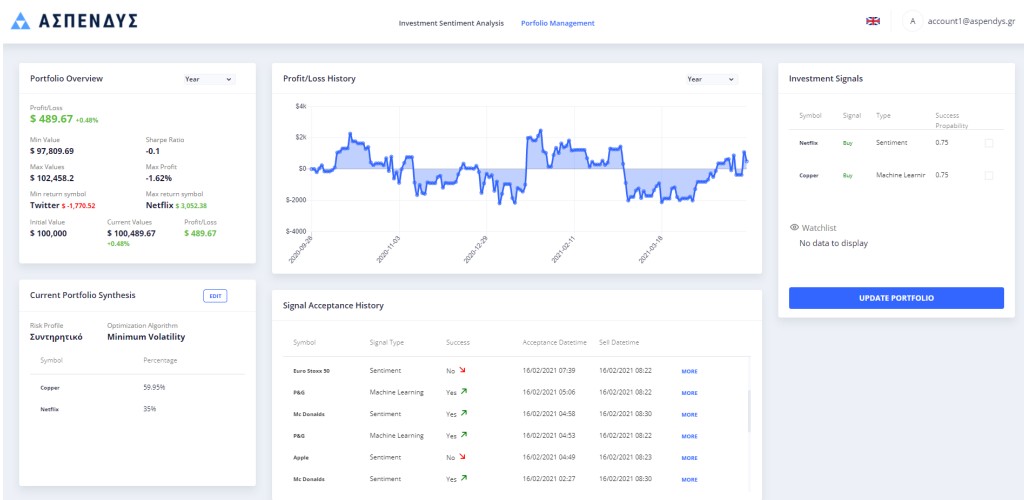

**Figure 2.** ASPENDYS Portfolio Management.

- The Portfolio Overview widget contains information regarding the user portfolio and their values are updated each time the platform receives news asset values. Specifically, the information illustrated contains:
    - Profit/Loss: The profit or the loss of the portfolio in both absolute value (dollars) and percentage.
    - Min Value: The minimum value in dollars that the portfolio has reached in the selected date period.
    - Max Value: The maximum value in dollars that the portfolio has reached in the selected date period.
    - Sharpe Ratio: The Sharpe Ratio value of the portfolio.
    - Min Return Symbol: The symbol with the minimum return as well as the absolute value in dollars of the return.
    - Max Return Symbol: The symbol with the maximum return as well as the absolute value in dollars of the return.
    - Initial Value: The initial investment value; by default, we have set this value to $100,000 for each user.
    - Current Value: The current value of the portfolio, which is the addition of both cash and assets value.
- The Profit/Loss History widget contains a line chart with the time series of profit/loss in the selected time period. The available periods are the current week, month and year.
- The Signal Acceptance History widget contains a table with all the BUY signals that have been accepted by the user. This table consists of:
    - Symbol: The name of the symbol to which the signal refers.
    - Type: The type of the signal depending on the generator that produced it (sentiment, technical, machine learning or mixed)
    - Success: A Boolean value that declares if the portfolio gain value is following this signal or not.
    - Acceptance Date-Time: The date and time that the user accepted this BUY signal for the specific asset.
    - Sell Date-Time: The date and time that the user accepted a SELL signal for the specific asset.

- The Current Portfolio Synthesis widget contains the percentages of all the assets of the portfolio as well as the selected Risk Profile and the Optimization Method. Moreover, in this widget, the user is able to edit their portfolio, as shown in Figure 3, by adding new assets, increasing their investment percentages or selling assets. Additionally, the user is able to change the Risk Profile or the Optimization Method, which are inputs to the portfolio optimization service.

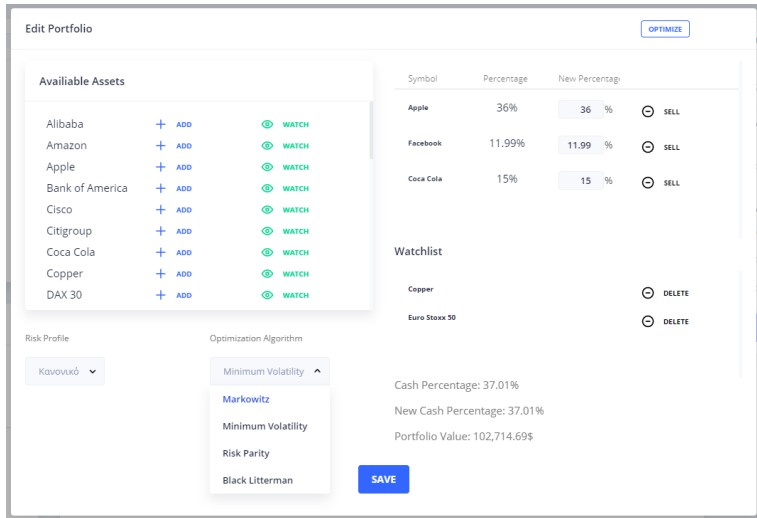

**Figure 3.** Edit User Portfolio.

### 4.1.2. Investment Sentiment Analysis

The Investment Sentiment Analysis view is one of the two main views of the AS-PENDYS platform. It contains unique widgets that assist the user to be informed about the sentiment of their portfolio assets, as well as the sentiment of articles that refer to these assets. More specifically, the Portfolio Investment Sentiment, the Investment Sentiment Change Notifications and the Sentiment Analysis of News and Articles are the three widgets that are included in the specific view, as shown in Figure 4 and analyzed below. The Investment Sentiment Analysis view also contains the widget Investment Signals that is common in both views of the platform. The specific view is fed by data that are generated by the components analyzed in Section 3.4.

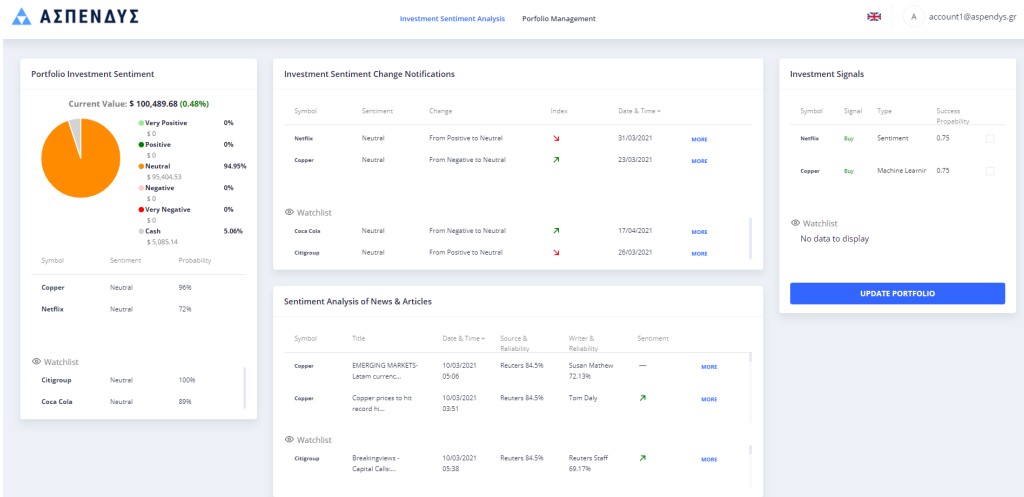

**Figure 4.** ASPENDYS Investment Sentiment Analysis.

- The Portfolio Investment Sentiment widget contains a pie chart with the percentages per sentiment for all the portfolio assets as well as the absolute value in dollars that corresponds to each sentiment. Moreover, this widget contains the current portfolio value as well as the percentage of profit or loss respectively. Additionally, in this widget, all the portfolio assets are displayed along with the state of their sentiment and the probability that this sentiment is accurate.
- The Investment Sentiment Change Notifications widget contains a table with all sentiment changes of the portfolio assets. The columns of this table are:
  - Symbol: The name of the symbol to which this entry refers.
  - Sentiment: The sentiment status of the specific symbol. The values it can be assigned are Very Negative, Negative, Neutral, Positive and Very Positive.
  - Change: The combination of the previous with the current state of sentiment.
  - Index: The index shows how much a symbol's sentiment state has improved or worsened.
  - Date/Time: The date and time describe when this change was detected.

  By selecting one of these changes, the user is able to see the timeline of the asset values together with the produced signals from the components in Section 3.7, as shown in Figure 5.

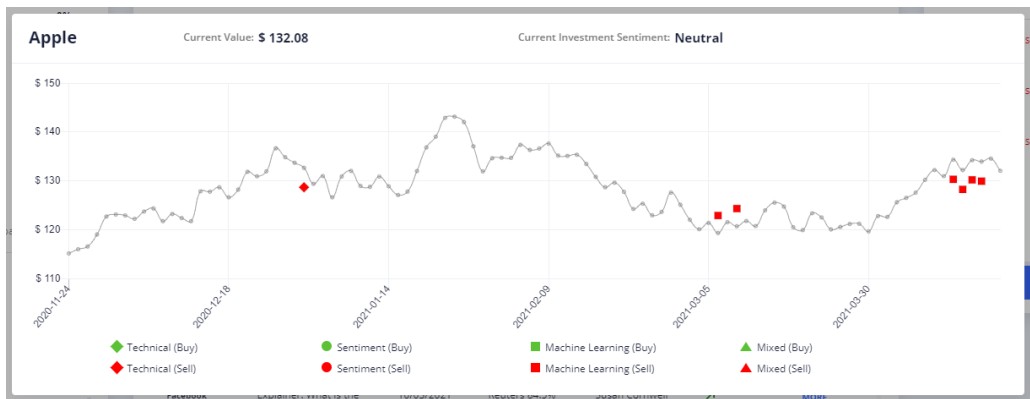

**Figure 5.** Asset values and signals timeline.

- The Sentiment Analysis of News and Articles widget contains the most recent articles and news that correspond to the assets of the users portfolio. The sentiment of each article is generated by the components analyzed in Section 3.4.1.

### 4.1.3. Investment Signals

The investment signals widget is a common widget in both Portfolio Management View and Investment Sentiment Analysis View. The data used in this widget are generated by the components in Section 3.7. As shown in Figure 6, it contains a table with the most recently generated signals for the assets that belong to the user portfolio.

The columns of the investment signals table include the symbol, signal, type and success probability, analytically:

- Symbol: The name of the asset that the signal refers to.
- Signal: The signal value can be either SELL or BUY.
- Type: The type of signal that can be Technical, Sentiment, Machine Learning or Mixed and indicates the generator that produced this signals Technical Analysis Generator, Sentiment Analysis Generator, Machine Learning Generator and Mixed Signals Generator, respectively.
- Success Probability: This probability is a metric that generated by each component and indicates the probability of this signal being successful.

The user is able to select one or more signals that they want to accept, and, by pressing the update portfolio button shown in Figure 6, the view in Figure 7 appears, where the

user is able to increase the percentage of the assets that have BUY signals or sell the assets with SELL signals.

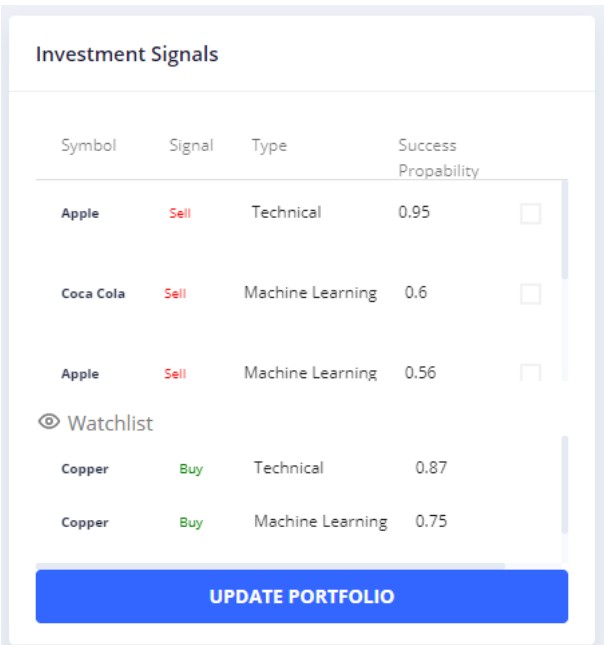

**Figure 6.** ASPENDYS Investment Signals.

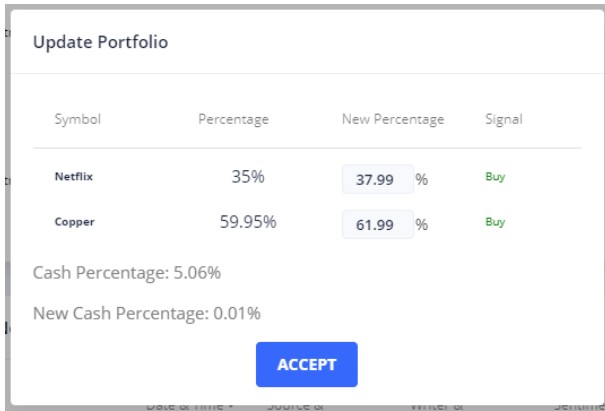

**Figure 7.** Update Portfolio using signals.

### 4.2. Application Use Cases

In this section, we analyze two different use cases of the ASPENDYS platform. The first use case is about the portfolio of a user that has invested a large amount of money into specific assets that have large value per piece using an aggressive risk profile, while the second use case is about a portfolio that has invested a smaller amount of money to assets with smaller value using a conservative risk profile.

More specifically, in Table 1, the assets of the aggressive portfolio are depicted as well as the amount that has been invested in each of them when they were initialized on 01/02/2021. The total amount of the investment is $79,833.25 and the remaining cash is $20,166.75.

Regarding the aggressive portfolio, on 05/02/2021, the sentiment analysis generator produced a BUY signal for the asset Twitter and the investor of this portfolio accepted this signal increasing the percentage of this asset to 35.22%. On 16/02/2021, the Technical Analysis Generator produced a SELL signal for the assets Facebook and Alibaba and the investor of this portfolio accepted this signal. Additionally, on 01/03/2021, the Mixed Generator produced a SELL signal for the asset Netflix and the investor of this portfolio

accepted this signal. The result of this use case was to increase the value of the portfolio by 2.95%, while, if the investor did not accept any of the signals, the value of the portfolio would increase by 0.69%. Figure 8 shows the progression of the portfolio value. The green line shows the actual value of the portfolio with the signals that have been accepted, while the blue line shows the portfolio value if the signals had been discarded.

**Table 1.** The assets of the aggressive portfolio, together with the invested amount in dollars as well as the percentage that each asset takes up in the portfolio.

| Asset | Percentage | Amount |
| --- | --- | --- |
| Facebook | 22.97% | $22,974.64 |
| Alibaba | 18.99% | $18,994 |
| Twitter | 16.89% | $16,891.3 |
| Netflix | 20.97% | $20,973.24 |

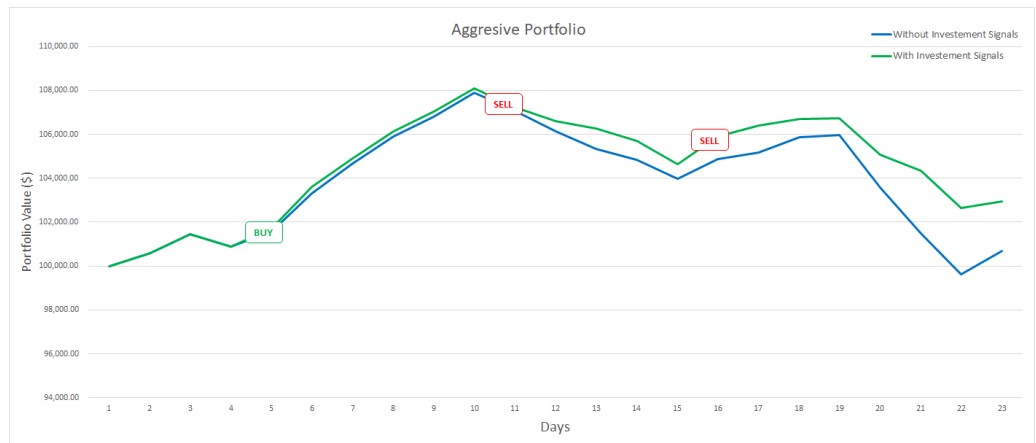

**Figure 8.** Aggressive portfolio value timeline.

In Table 2, the assets of the conservative portfolio are illustrated as well as the amount that has been invested in each of them when they were initialized on 01/02/2021. The total amount of the investment is $26,534.11 and the remained cash $73,465.89.

**Table 2.** The assets of the conservative portfolio, together with the invested amount in dollars as well as the percentage that each asset takes up in portfolio.

| Asset | Percentage | Amount |
| --- | --- | --- |
| Google | 5.86% | $5863.11 |
| Apple | 6.11% | $6119.84 |
| Oracle | 4.25% | $4259.61 |
| Tesla | 8.09% | $8091.89 |
| Cisco | 2.19% | $2199.66 |

Regarding the conservative portfolio, on 06/02/2021, the Machine Learning Generator produced a SELL signal for the asset Apple and the sentiment analysis generator produced a SELL signal for the asset Tesla, the investor of this portfolio accepted these signals. On 22/02/2021, the Technical Analysis Generator produced a BUY signal for the asset Oracle and the investor of this portfolio accepted this signal increasing the percentage of this asset to 18.32%. The result of this use case was to increase the value of the portfolio by 3.28%, while, if the investor did not accept any of the signals, the value of the portfolio would have decreased by 2.14%. Figure 9 shows the progression of the portfolio value. The green line shows the actual value of the portfolio with the signals that have been accepted, while the red line shows the portfolio value if the signals had been discarded.

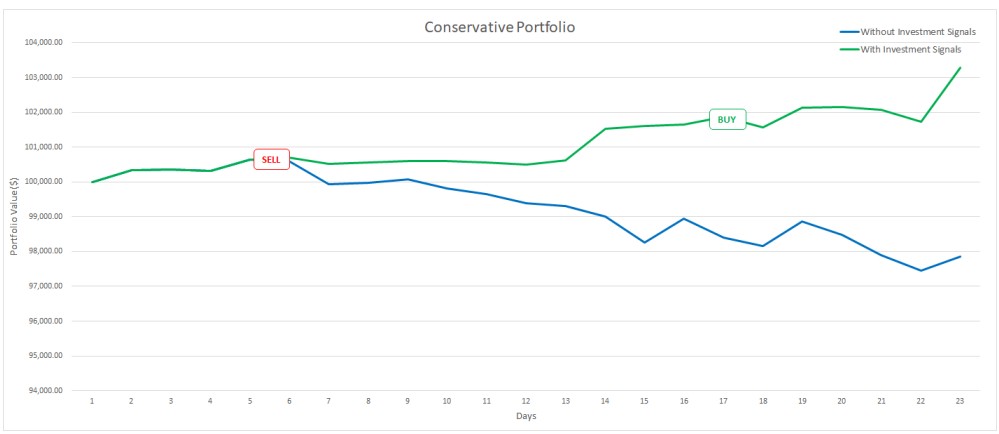

**Figure 9.** Conservative portfolio value timeline.

## 5. Discussion

In recent years, the revolution in data analysis offered by the big data approach enables the generation of new AI tools in several fields. These advances have created the need for new decision-making systems for the generation of modern investment models. The current trends in the financial sector create the demand for automated portfolio management services. This type of portfolio management can combine several approaches such as machine learning using data analysis to select underlying investments and technical analysis systems assessing the likely magnitude of potential price moves based on historical correlations. Moreover, nowadays, opinions are no longer something that needs to be sampled via focus groups or surveys. Instead, content derived from social, event and news media sites and vendors who interpret and spread those data are providing data streams that offer insight into the thinking of their myriads users. The sentiment analysis of news stories and social posts using machine learning is a trending topic. There are trained models that automatically identify if social media posts and news stories about an asset are positive or negative, assisting investors to stay ahead of future price movements.

In this paper, we describe in detail the ASPENDYS platform, an AI-enabled platform for financial management. More specifically, the ASPENDYS platform is a high-level interactive interface offering end users the functionalities of monitoring, modifying and expanding their portfolio. The AI technologies that were developed in the context of this project produce investment signals through the components of the ASPENDYS system such as machine learning, sentiment analysis and technical analysis component. This creates a dynamic tool that can be used by investors in order to assist them in the decision-making process. The user interface is a modern web application that combines all the capabilities of a Web 3.0 application, processing information with near-human-like intelligence, through the power of AI systems, to assist end users in their investment decisions. Different data sources were used in this project to extract information regarding the assets that we studied. Specifically, social media posts from Twitter and Stocktwits as well as articles from news agencies such as BBC News, Reuters, Coin-telegraph, The Wall Street Journal and Bloomberg were used for the production of the assets' sentiment. Moreover, the Yahoo! Finance API was utilized to acquire the assets' daily values for the technical analysis and machine learning signals prediction.

During the use cases examined in this paper, we concluded that the use of the AS-PENDYS platform in the investment decisions process could increase the profit of a portfolio. More specifically, in the aggressive portfolio use case, the ASPENDYS platform suggestions assisted the investor to increase their portfolio value by approximately 3%, while, in the use case of the conservative portfolio, the suggestions led to an increase of almost 5% in the portfolio value. The technologies used in the specific project implement state-of-the-art algorithms in the area of technical analysis, machine learning and sentiment analysis, a necessary feature in a modern portfolio management and model-based trading platform. In addition, the combination of all these technologies in the field of stock

market prediction positions the ASPENDYS platform in the Web 3.0 applications of the financial sector.

## 6. Conclusions

As mentioned in Section 2, there are several works that deal with stock prediction; most of them are based on stocks' historical values, without taking into consideration information that may affect the stock prices, such as social or political events. While referring to RQ1, the system takes into account both financial data (e.g., stocks' closing values) and textual data retrieved from either reputable news websites or social networks (i.e., Twitter and Stocktwits). For processing the financial data, different methods of technical analysis and machine learning [5,50] are utilized. The outcome of the analysis is the generation of investment signals for buying or selling stocks. Because the textual data are retrieved from various sources and in large quantities, they are firstly correlated through text analysis with the available financial symbols of the platform, and then filtered based on their reliability. Then, only the reliable data are analyzed for extracting their sentiment. By aggregating and recording the daily sentiment of each stock, time series with the sentiment of the stocks are created. By combining these time series and the price indices of the stocks, the system is able to recommend additional signals based on the general sentiment for the stocks.

Additionally, regarding RQ2, the ASPENDYS platform can serve users as a mean for managing and monitoring their investments. After being registered on the platform, the users can define a set of assets that they are interested in investing into or monitoring their course. This set of assets forms the user's portfolio. The system is capable of predicting investment signals, related to a portfolio, and alerting the end users when necessary. Finally, the platform provides, through dedicated tools, the complete analysis of the user's investment movements, presenting historical data with their investments, their portfolio profit within a certain period and the course of each asset independently.

However, there are some limitations in the proposed platform. More specifically, the fine tuning of sentiment analysis models is a difficult task because there is a lack of open source annotated datasets that contain articles and tweets in the field of economics. Therefore, as mentioned in Section 3.7.1, we have to use datasets from another sector that is a sub-optimal option. From a technical point of view, the specific implementation has not been tested in terms of scalability; however, a future plan is the deployment of the ASPENDYS platform in a business infrastructure where these kind of tests can be done. Moreover, another future plan for the proposed platform could involve the implementation and the adaptation of the algorithms and models to the cryptocurrency industry. In addition, the part of the sentiment analysis could be enhanced in the future by integrating lexicons such as the VADER (Valence Aware Dictionary for sEntiment Reasoning) lexicon [61].

**Author Contributions:** Conceptualization, A.D., A.K. (Athanasios Konstantinidis) and G.P.; methodology, T.-I.T., A.Z. and A.K. (Athanasios Konstantinidis); software, T.-I.T., A.Z. and T.P.; validation, M.S., A.K. (Anna Kougioumtzidou), K.T. and D.P.; formal analysis, G.P.; investigation, M.S. and A.K. (Anna Kougioumtzidou); resources, K.T. and D.P.; data curation, T.-I.T. and A.Z.; writing—original draft preparation, T.-I.T. and A.Z.; writing—review and editing, T.-I.T. and A.Z.; visualization, T.-I.T. and A.Z.; supervision, A.D.; project administration, D.T.; and funding acquisition, A.D. All authors have read and agreed to the published version of the manuscript.

**Funding:** This work was partially supported by the ASPENDYS project co-financed by the European Regional Development Fund of the European Union and Greek national funds through the Operational Program Competitiveness, Entrepreneurship and Innovation, under the call RE-SEARCH–CREATE–INNOVATE (project code:T1EDK-02264).

**Data Availability Statement:** All data that are not subjected to institutional restrictions are available through the links provided within the manuscript.

**Conflicts of Interest:** The authors declare no conflict of interest.

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
