# Peer review of "An AI-Enabled Stock Prediction Platform Combining News and Social Sensing with Financial Statements"

_futureinternet, doi:10.3390/fi13060138_

Round 1

Reviewer 1 Report

I congratulate the authors on their work. I have a few questions on issues that are not entirely clear to me - they can also be taken as suggestions for improvements in the text and the tool itself:
1. are the authors familiar with and have they used the study "Twitter mood predicts the stock market" Authors: Johan Bollen, Huina Mao, Xiaojun Zeng https://doi.org/10.1016/j.jocs.2010.12.007 and the mood tracking tools used there, namely OpinionFinder that measures positive vs. negative mood and Google-Profile of Mood States (GPOMS)? Do they see an opportunity to additionally implement these tools in their project?
2) There are many fake accounts on Twitter. Astroturfing is also a known phenomenon. Could this affect the reliability of the data obtained? How did the authors protect against this? Is the technique described in point 2.4 sufficient to avoid these risks?
3) Did the authors use dictionaries available on the Web (SentiWordNet or Hu and Liu's lexicon) for asset sentiment? If not - why not? In which languages are the analysed tweets downloaded? Only in English, or in other languages?
4) Does the algorithm take into account other risk factors, such as the global political situation or events like the blockade of the Suez Canal and their possible impact on the markets?

Reviewer 2 Report

Dear Authors,

Thank you for submitting your paper to “Future Internet”. I read it with great interest, and I am happy to admit that it can be published after some revisions. First of all, your article offers a significant contribution to both theoretical and practical studies on stock prediction. Second of all, it is based on multi-threaded and interdisciplinary research with very interesting results which could be analysed (i.e. verified or discussed) in other academic papers. However, your paper needs to be polished because it has several flaws. I ask you to revise your article to improve its quality. Below you will find my greatest concerns – they are listed along with recommendations of possible improvements which I would like you to consider.

The major problems are the structural and theoretical aspects of your work. For example, in the “Introduction”, you could add the reference to the primary aim of your study and describe the structure of the paper. The study is very complex, so at this stage, it would be helpful to characterise the organisation of the text and your narrative intention (you could move here the text from lines 177-187 and enrich them with the proper references to other chapters). It would also be great to add a source to the statement given in 36-38 lines (it is very general now). 

The “Related work” is, in fact, a literature review. I would like to ask you for moving this part to a separate chapter (number 2). This part is presented chaotically. The topics are mixed and not entirely understandable. The content needs to be clarified to be coherent. I suggest introducing a subchapter that could organise your work. For example, there could be a literature review regarding the “before-AI” stock predictions solutions (including the news and social sensing). Then you could present state of art describing the AI-era of stock predictions (of course, with the inclusion of the news and social sensing). You could also define AI phenomena in one-two sentences (it would improve the scientific soundness of your paper). As a result of your literature review, you should identify specific academic gaps (i.e. diagnose what has not yet been researched in the adopted research are; what constitutes a niche in research on the problem you have identified). And on this basis, you could formulate research questions that your paper does not have yet. As a result, section 1.2 could be named "Defining the academic gaps and research questions". 

The current form of part 1.2 is not clear to me. First, you did not convincingly prove that "the combination of machine learning, sentiment analysis and the financial stock prices prediction is a hot topic" (the literature review part could be enriched to confirm that). On the other hand, you did not justify the need for a new model or its uniqueness (which should also be apparent from the literature review). In other words, the 1.2. the part should robustly demonstrate the need for implementing your model as a response to identified research gaps. 

I also think that 156-175 lines are more appropriate for the “Conclusions” part. Right now, describe how the platform works before the proper description in part 2 (I would say that we read the conclusion, a kind of summary, before knowing the research process).

Additionally, it would be good to define the following terms shortly: Web 3.0IR,  NLP and DL. It also would be better to avoid colloquialisms such as „hot topic”. 

Part 2 is problematic. I would like to ask you for presenting the methodology before you start describing the model itself. You could place it in a separate subchapter with information about the research methods in your project and your paper. Was it observatory participation, a case study, an experiment? When did the work begin? How did it progress in stages? Who took part in them? How was the model validation process carried out? What were the criteria of choosing, i.e. Twitter and Yahoo, for analysis? In other words, the methods part should focus only on methodology, the research process description, the road you have travelled together to develop such an elaborate model. Then the “System Architecture” could follow.

The model presentation is fascinating, and I highly appreciate it. It offers a brand new idea of your authorship, and its construction could be discussed and verified by other researchers. I read it with great interest. I would like to ask you for adding proper sources to the 424-425 lines and then to 426-444 lines (it would enhance the scientific soundness). I also liked the “Results” part.

In the “Discussion”, I found a part that would fit better to the methodological subchapter. The lines 681-694 refer to the research process and offer new information about it (it would great if you explain why you chose the IMDb).

Unfortunately, your paper lacks the “Conclusions” part. I strongly suggest adding this at the end of your text regarding research goals and questions. It would help if you enriched it with the limitations of your study and ideas for future research on your model. It would be interesting to read about the contribution of your work to the theory and practice in the adopted research field.

The last thing is language – I am not a native speaker, but I have noticed some misspellings, edition and grammar mistakes which should be revised (i.e. lines 218, 654).

Dear Authors, I hope you will consider my suggestions and decide to implement them. I am looking for improving your work because it is very interesting and based on reliable research.

Sincerely.

Reviewer 3 Report

Thank you for the opportunity to review the paper "An AI-enabled Stock Prediction Platform combining News and Social Sensing with Financial Statements". The paper addresses an interesting and novel theme, about the emergence of huge data volumes (big data) and the advent of more powerful modelling techniques such as Deep Learning, by presenting the reader a supportive platform for investors that combines various methods from the different scientific fields, as part of a bigger ASPENDYS project.

In regards to this topic, the advent of big data and machine learning in Web 3.0 technologies, the financial forecasting and the model-based trading has gained a growing interest, as the authors suggest.   

Overall, the subject is correct attributed to this journal in regards to the aim of the study, and the section regarding the presentation of the ASPENDYS platform is detailed and very well supported by in-depth descriptions, examples, figures etc. Also, another interesting point pf the article is that the authors present in comparison others financial forecasting and portfolio management platforms utilizing AI technologies.

However, there are some issues that should be addressed in this revision. First of all, I recommend the authors to take into consideration some corrections about the structure and the content of the article:

  1. The way are introduced the works of different scholars in citations must be improved so the general flow of reading is not interrupted by expressions as: In study [5] (Line 64), Authors in work [7] (Line 68), In [8] (Line 74), In the work of [13] (Line 89)… and so on.
  2. Add a Conclusion section.
